# COVID-19 Drug Repurposing: A Network-Based Framework for Exploring Biomedical Literature and Clinical Trials for Possible Treatments

**DOI:** 10.3390/pharmaceutics14030567

**Published:** 2022-03-04

**Authors:** Ahmed Abdeen Hamed, Tamer E. Fandy, Karolina L. Tkaczuk, Karin Verspoor, Byung Suk Lee

**Affiliations:** 1School of Cybersecurity, Data Science and Computing, Norwich University, Northfield, VT 05663, USA; 2Sano Centre for Computational Medicine, 30-072 Kraków, Poland; k.tkaczuk@sanoscience.org; 3Department of Pharmaceutical and Administrative Sciences, University of Charleston, Charleston, WV 25304, USA; tamerfandy@ucwv.edu; 4School of Computing Technologies, RMIT University, Melbourne 3001, Australia; karin.verspoor@rmit.edu.au; 5School of Computing and Information Systems, The University of Melbourne, Melbourne 3010, Australia; 6Department of Computer Science, University of Vermont, Burlington, VT 05405, USA; bslee@uvm.edu

**Keywords:** Coronavirus pandemic, COVID-19 treatment, drug repurposing, adjuvant treatment, clinical trials ground-truth, drug association maps, literature mining, clique identification

## Abstract

Background: With the Coronavirus becoming a new reality of our world, global efforts continue to seek answers to many questions regarding the spread, variants, vaccinations, and medications. Particularly, with the emergence of several strains (e.g., Delta, Omicron), vaccines will need further development to offer complete protection against the new variants. It is critical to identify antiviral treatments while the development of vaccines continues. In this regard, the repurposing of already FDA-approved drugs remains a major effort. In this paper, we investigate the hypothesis that a combination of FDA-approved drugs may be considered as a candidate for COVID-19 treatment if (1) there exists an evidence in the COVID-19 biomedical literature that suggests such a combination, and (2) there is match in the clinical trials space that validates this drug combination. Methods: We present a computational framework that is designed for detecting drug combinations, using the following components (a) a Text-mining module: to extract drug names from the abstract section of the biomedical publications and the intervention/treatment sections of clinical trial records. (b) a network model constructed from the drug names and their associations, (c) a clique similarity algorithm to identify candidate drug treatments. Result and Conclusions: Our framework has identified treatments in the form of two, three, or four drug combinations (e.g., hydroxychloroquine, doxycycline, and azithromycin). The identifications of the various treatment candidates provided sufficient evidence that supports the trustworthiness of our hypothesis.

## 1. Introduction

### 1.1. Significance Statement

Though several COVID-19 vaccines are available, there is no guarantee that everyone will choose to be vaccinated. Moreover, with the virus constantly mutating and new strains are emerging (e.g., Delta [1], Omicron [2]), much research is needed to decide the efficay of the available vaccines against the new variants. There is a critical need for effective treatment, beyond prevention. Drug repurposing has already provided a clear path to COVID-19 antiviral treatment. For instance, the oral antiviral molnupiravir, which was originally FDA-approved for SARS-COV, now shows to reduce the risk of hospitalization or death by approximatly 50% in COVID-19 patients [3]. Recently, the ritonavir-boosted nirmatrelvir has been investigated as an antiviral treatment [4]. A recent clinical trial concluded that the two drugs combined have reduced the risk of hospitalization or death by 89% in COVID-19 patients, when compared to other treatments [5]. Ritonavir drug was among the top-five drugs from a list of 30 drug candidates we recommended as a coronaviruses treatment [6].

These previous findings have shown the significance of drug repurposing for identifying potential COVID-19 treatments. The evidence underpinning this was derived from a small set of publications that were originally published about the coronavirus that caused the Severe acute respiratory syndrome (SARS). SARS is an illness known to be caused by the SARS-associated coronavirus (SARS-CoV) that broke out prior to the SARS-CoV-2  [7]. Now that the scientific community has gained significant knowledge specficially about the COVID-19 disease, vaccines, and drug treatments, this creates an opportunity to directly investigate the massive (and growing) COVID-19 biomedical literature as a resource for identifying potential new opportunities for treatment. More interestingly, our previous research illuminated the importance of identifying drug combinations by means of text mining to identify such a treatment [6]. Here, we further argue that such a treatment may be identified from drugs that are highly associated, as long as the same associations hold in clinical trials.

### 1.2. Computational Drug Repurposing Background

There is no doubt that our world has never seen a greater public health ordeal in recent times than the Coronavirus pandemic. Such desperate times called for accelerated processes to avoid millions of infections and to help save the lives of those who were infected. While the process of vaccine development typically takes years to fully mature, its timeline was significantly compressed [8]. In parallel, the promise of drug repurposing was immediately investigated in hundreds of clinical trials. Drug repurposing enables acceleration of treatment development through identification of a new indication for an existing FDA-approved drug(s) [9,10,11].

Here, we motivate our work by listing and presenting the various computational approaches that have addressed drug repurposing: artificial intelligence (AI) including machine learning [12], biomedical literature mining and semantic knowledge representation, network-based drug repositioning, clinical analysis, signature matching, molecular docking, and experimental phenotypic screening [13]. One that stands out among many is AI. Various models particularly highlighted how to use AI for accelerating COVID-19 drug repurposing [14,15]. A recent review by Wang and Guan [16] grouped drug repurposing methods into three categories (computational research, clinical trials, and in vitro experimental studies). The study highlighted certain computational methods including network-based algorithms and expression-based algorithms. Wang [17] used another computational method to address the drug repurposing using a virtual docking screening of approved drugs, which was followed by a molecular dynamics simulations to find rational drug design targeting SARS-CoV-2. Gysi et al. [18] used anther rather complex method that combined AI, network diffusion, and proximity networks to investigate 6340 drugs to treat; the study concluded that 76 drugs have reduced viral infection that cannot be identified using docking-based strategies. Computational drug repurposing investigation also uses statistical analysis [19,20,21]. Further, clinical trials also have shown promise in providing insights into untested drugs [22].

Text mining is a computational method commonly used and frequently investigated for drug repositioning [23], as well as for more general applications relevant to COVID-19 [24]. Li et al. [25] constructed a molecular network that was seeded by features extracted from the biomedical abstracts published prior to the COVID-10 pandemic. This study resulted in the identification of 30 candidate drugs that are potentially repurposeable for COVID-19 treatment. The promise of biochemical knowledge for drug repurposing inspired the construction of COVID-19 ontology [26], capturing properties of chemical entities as well as virological, epidemiological, and clinical aspects of COVID-19. Another study by Kuusisto et al. [27] found that mining a large collection of biomedical publications using a word-embedding approach around FDA-approved drugs shows promise. Due to the similarity between SARS-CoV-2 and SARS-CoV, the study claimed that the treatments of SARS may also then be appropriate for COVID-19 (to the Coronaviridae family of the Nidovirales order [28]). At the early phase of the global pandemic, Baker et al. [29] investigated the potential repurposing of quaternary ammonium compounds. Text mining the biomedical literature was the tool of choice to investigate whether such a treatment would be appropriate for the coronavirus. In a more recent study, Muramatsu and Tanokura [30] presented a novel tool that investigated candidate COVID-19 drugs which was based on the relationships obtained from text mining of the vast literature in the form of biomedical abstracts. Tworowski el al. [31] combined features from publication references and clinical trials to develop a searchable COVID-19 drug repository that enables users to search for general information about FDA-approved drugs, recipes, and the drugs’ mechanisms of action.

Machine learning (ML) has also played a significant role in the identification of virus targets and molecular docking in COVID-19. Deep learning and deep docking were amnong the methods have been heavily investigated [32]. Artificial neural network (ANN) models are particularly useful to train with virus protein sequences as inputs and antiviral agents are deemed safe in humans as outputs [33]. ML methods also reveal the relationship between viral, drug and the host proteins [34]. Traditional ML approaches also played an important role in the COVID-19 drug repurposing. A supervised model (linear support vector machine) was used to classify COVID-19 patients into two categories: (1) those with non-severe COVID-19 symptoms, and (2) those with severe symptoms. The model used a number of candidate proteins and linked them to two FDA-approved drugs, namely, ponatinib and selinexor for potential repositioning [35]. Both unsupervised and supervised ML can be used to cluster and classify FDA-approved drugs based on their mechanism of action (MoA). This approach established the foundations of drug repurposing and also provided insights to discovery of MoAs of new drugs in COVID-19 [36].

The significance of computational methods in COVID-19 drug repurposing is undeniable. Our previous work on this subject is based on similar foundations, in terms of both the data and methods. Specifically, our previous work examined the SARS-CoV-2 biomedical literature [6] to identify both drugs and protein targets. Further, a map was constructed and used as a computational data model for further investigations. The study illuminated an important finding that in some cases it may be necessary to use more than one drug in the course of COVID-19 treatment. This insight has inspired the further investigation in this current work.

There are substantial differences in this work from our previous work. First, due to the novelty of COVID-19 at the time of the previous work and the availaibility of only a small number of publications, the similarity between SARS-CoV and SAR-CoV-2 suggested the investigation of SARS-CoV biomedical literature to gain insights. Now, due to the abundance of SAR-CoV-2 research publications and the wealth of knowledge within, we directly investigate them for further insights. Second, this work is focused only on the identification of drug names as possible treatments. No other entity or target (e.g., gene or protein) is the concern of this work. Particularly, we study strongly-associated drugs that were mentioned in the biomedical literature. We further validate such associations from evidence extracted from clinical trials that are designed to study the same drugs. The evidences from the two sources are compared for similarity using a novel algorithm, which we present here. The intersection of the two results presents possible candidates for COVID-19 treatment. In our approach, we focus exclusively on associations forming a special map structure know as a “clique”. This ensures that the drugs identified from both sources are strongly connected within the source literature; this may be conceived as reliable evidence for further testing.

The rest of this paper discusses the following steps of our method: (1) how the drug names were identified from both publications and clinical trials using the Chemical Entities of Biological Interest (ChEBI) ontology, (2) how the drug maps are constructed from the biomedical literature and the clinical trials, respectively, (3) how the maps are pruned to remove weak associations, and (4) how strongly-connected drugs are identified in the map and tested for similarities until COVID-19 treatment candidates emerge.

## 2. Materials and Methods

### 2.1. Datasets and Resources

The experiments and findings of this work were derived from two distinct datasets extracted from the PubMed and ClinicalTrials.gov web portals, respectively, using the search query “COVID-19”. The cutoff date was 25 June 2021. Any publication past this date is not part of this study. As a result, a set of 110,000 COVID-19 related publications and a set of COVID-19 clinical trial records were extracted. The publications were analyzed for drug names and drug associations; and the clinical trial records (intervention and treatment sections) were used to validate the drug associations identified in the publications.

### 2.2. Computational Approaches

The methods of this work require the following steps: (1) drug name extraction, (2) association analysis and network construction, (3) clique detection, and (4) validation against clinical trials. Figure 1 shows the workflow steps to generate the candidate treatments.

#### 2.2.1. Drug Name Extraction

The biomedical publications dataset we used for the analysis is composed of a set of text documents. Each document is described using two key fields: ID and abstract. The ID captures the identification of the publication in the digital repository. This identifier is necessary to link back to the original publication during the validation process. The abstract is a plain text describing the research and findings of each publication, and contains the names of such entities as diseases, symptoms, drugs, and any others relevant to the study topic of the publication. Identifying such entities in an abstract requires text processing, which includes parsing text, removing noise words, and the actual indentation of entities. Here, we use a *dictionary-based drug name extraction*, using the chemical terms in the ChEBI [37] ontology. This approach has demonstrated efficacy for identifying ChEBI terms [38]. The abstract text is checked against the ontology terms. When there is a match between a token and a ChEBI drug name, we extract the term and store it with its ontology term ID. This contributes a record that links the drug name and the ChEBI term ID to the publication. This also guarantees that the terms co-occurring within the same publication are linked together. This sets the stage to analyze the association among the drugs that are mentioned in the same publication, and across the entire dataset.

#### 2.2.2. Association Analysis and Network Construction

The identification of drug names and the co-occurrences in publications naturally provide a rich map that can be used for further investigations. However, the identification of every chemical entity (that is not a drug) may become a source of noise. For this reason, we have applied an association analysis algorithm that filters out the noise. Apriori [39] is one of the most prominent algorithms and also well-known for association rule learning. The algorithm must be configured using a “support” parameter as a base threshold, and, therefore, provides a certain level of noise-removal to keep only the associations that meet the threshold.

Since the drug co-occurrence records (from the biomedical literature datasets) are typically too many, we chunk the dataset into subsets of 7000 publications. This helps to speed up the Apriori algorithm to compute associations between the drugs. Such associations present important links since they lend themselves to a network model, a map that we use in the next step. When the drug links are combined based their associations they naturally form an undirected network model. The outcome is a map that provides the foundation for further discovery.

#### 2.2.3. Clique Detection

As stated above, the ultimate purpose of this research is to identify communities of drugs that are working together to provide COVID-19 treatment. In the graph theory, a clique is a small set of nodes in a network such that every two distinct nodes in the clique are adjacent, hence, directly connected via a link. Therefore, a clique guarantees the interaction among all constituents and corresponds directly to the notion of community that we are after.

After the network construction step above is completed, we load it into the networkX [40] framework and then apply the maximal clique algorithm [41] to identify the cliques. Various experiments we conduted identified cliques of size two, three, and four. This step marks the end of the literature mining phase.

#### 2.2.4. Clinical Trials Analysis

We apply the same steps (drug name extraction, association analysis and network construction, and clique detection) to texts in the intervention and treatment sections of the collected COVID-19 clinical trial records (Section 2.1). This results in clinical trial cliques that we used to validate the literature cliques.

#### 2.2.5. Validation and Discovery

We use the clinical trials as a validating tool to the findings from the medical publications. The intuition underlying this approach is that clinical trials contain drug combinations that are already under investigation and, therefore, represent combinations with a strong underpinning hypothesis that their combination is meaningful and effective. We therefore compare the emergence of the same size cliques (and their constituents) in the network constructed from the drugs mentioned in the literature, with those identified in clinical trials.

Methodologically, the map of drugs extracted from medical publications and the combination of drugs identified from the literature are validated against the corresponding information extracted from the clinical trial records. We compared cliques detected from publications with cliques detected from clinical trials under the condition of being of the same size. If a match is found (i.e., a clique is supported by both the clinical trials and the publications) then we consider it a possible candidate for further investigations, otherwise, it is ignored.

The steps of this algorithm, called *Search-n-Match*, are defined in Algorithm 1. The algorithm is executed in a step-by-step fashion for each clique in the literature-derived set, until all candidate drug combinations are produced.
**Algorithm 1** Search-n-Match.1Let P and Q be two cliques detected from biomedical publications and clinical trial records, respectively.2The condition that both cliques must have the same length is checked before the search begins.3If not of the same length, the process is rejected, otherwise it continues.4The components of the clique P are compared one at a time with each of the components in the clique Q.5If there is a match, the matching components are stored in the result set.6The process is repeated until all components of the clique P are compared with all components of the clique Q.7The similarty score is calculated using a modified version on Jaquard’s similarity [42] index, which calculates the size of the intersection divided by the size of of the union. Because we apply the restriction of comparing cliques that have the same size, we divide intersection by the size of either clique.8Upon completion, the algorithm terminates and intersection of the two cliques is returned along with the similarity score.

## 3. Results

Due to the large number of publications analyzed, the drug-mention cardinality was also high. The association analysis step pruned some association links. Cliques of size two to six were observed in the network constructed from the medical publications dataset. On the other hand, in the network derived from the clinical trials, only cliques of size two to four were observed. Applying the constraint of investigating only cliques of the same size in the two networks, cliques of size five to six derived from the literature dataset were not investigated. Table 1 shows the statistics of analyzing the biomedical publication networks for cliques. The rows shows the clique size for each fold of 10 folds. The colums show the numbers of cliques generated from each fold. We compared the cliques of the same size with the cliques resulting from the clinical trial records. The clinical trials were much smaller in size (5578 records), and was analyzed in its entirety as one fold. The cliques returned for each size are as follows: 78 clques of size two, 10 cliques of size three, and 6 cliques of size four. Clearly, the very small number of cliques found in the network contructed from the clinical trial records (92 cliques) eliminated 3551 cliques detected from the publications. This shows the significance of using the clinical trials records as a validation mechanism. The comparison identified the matches of maximum three of the four components that constructed each clique. This analysis suggested that the COVID-19 treatment candidates can be made of at most three different drugs.

Table 2, Table 3 and Table 4 present the drug cliques identified in the clinical trials data. Figure 2 shows the chemical entities and drugs extracted from the publications on one hand and clinical trials on the other. Clearly there is an overlap between the results from the two sources, which supports the validation methods we are presenting here.

## 4. Discussion

### 4.1. Two-Drug Combinations

We discuss the various findings of this work starting with two-drug combinations. Table 5 lists drug pairs resulting running the algorithm and validating the literature finding against clinical trials. This result constitues grounds for whether such a combination is possible to combine. Note that being possible does not mean recommended until validated by human experts. The “*combineability*” of each pair listed in Table 5—the plausibility of the combination based on biochemical and pharmacological properties—is discussed below.

Estrogen (hormone) and estradiol (hormone) are not possible to combine. Estradiol is structurally identical (bioidentical) to estrogen produced in ovaries. Estradiol is one form of estrogen—there are others, too—and may be administered by a number of routes (e.g., by mouth, through the skin). It would not make sense to combine these two drugs together [43,44,45,46,47,48,49].Hydroxyethylidene, is etidronic acid, known as a drug its generic name is etidronate and azithromycin (macrolide antibiotic) are possible to combine. Azithromycin is an antibiotic (working against bacterial infections) that also has antiviral and anti-inflammatory properties. We found 1-hydroxyethylidene-1 listed as a synonym for Etidronic acid [50], the first generation bisphosphonate. Etidronate has been discontinued in the US though there are no drug interactions [51,52,53].Lopinavir (protease inhibitor) and ritonavir (protease inhibitor) are possible to combine. In fact, this combination already exists as an FDA-approved medication under the brand name Kaletra [54,55,56,57,58].Ruxolitinib (janus kinase inhibitor) and Colchicine (anti-gout) are possible to combine. Ruxolitinib (as the systemic treatment) is used for myelofibrosis (bone marrow cancer), polycythemia vera (a type of blood cancer), and graft-versus-host disease (a complication of bone marrow transplant). Colchicine is a medication to prevent and treat gout (too much uric acid). There are no reported drug interactions with this combination [59,60,61,62].Hydroxychloroquine (antimalarial) and favipiravir (antiviral) are possible to combine. Hydroxychloroquine is a medication used to treat malaria. Favipiravir is an antiviral developed for treating influenza [63]. It is not commercially available in the US. There are apparent drug interactions with this combination, although data are limited as favipiravir is not available [64,65,66,67].Hydroxychloroquine (antimalarial) and chloroquine (antimalarial) are not possible to combine. There is a known major drug interaction between them—increases QT-interval prolongation (causing irregular heartbeats). Hydroxychloroquine is an analog to chloroquine. They are different drugs, but essentially do the same thing. Clinically it would not make sense to combine them.Azithromycin (macrolide antibiotic) and ivermectin (anthelmintic) are possible to combine. Azithromycin is an antibiotic (working against bacterial infections) that also has antiviral and anti-inflammatory properties. Ivermectin is an antiparasitic (working against parasite infections). There are no known drug interactions with this combination.Hydroxychloroquine (antimalarial) and lopinavir (protease inhibitor) are probably not possible to combine. Hydroxychloroquine is a medication used to treat malaria. Lopinavir is a protease inhibitor used in the management of HIV. the UpToDate database [68] does not list major drug interactions. Unon searching Micromedex database [69] it indicated that there is a drug-drug interaction – the combination leads to a prolongation of QT interval.Hydroxychloroquine (antimalarial) and doxycycline (tetracycline antibiotic) are possible to combine. Hydroxychloroquine is a medication used to treat malaria. Doxycycline is an antibiotic used to treat bacterial infections. There are no reported drug interactions with this combination.Daclatasvir (antihepaciviral) and sofosbuvir (nonstructural protein 5B (NS5B) nucleoside polymerase inhibitor) are possible to combine, and their combination is already used in the treatment of hepatitis C. Daclatasvir is not available in the US, but the combination is marketed under the brand name Darvoni in other countries. There is a minor drug interaction; Daclatasvir may increase the concentration (the level in the body) of sofosbuvir.

### 4.2. Three or More Drugs Combination

As indicated in the results in Section 3, combinations of three and four drugs have also emerged. To identify a possible treatment, we ran the Match-n-Search algorithm against those combinations, identified their similarities, and extracted the common combinations. To demonstrate this step, we depicted a network in Figure 3 that displays two sample cliques of size four: (1) a clique extracted from literature, color-coded in green; and comprised of (hydroxychloroquine, darunavir, lopinavir, and favipiravir), (2) a clique extracted from the clinical trial records, color-coded in red; and comprised of (hydroxychloroquine, lopinavir, ritonavir, and favipiravir). By applying the Search-n-Match algorithm it produced the following three-drugs combination: (hydroxychloroquine, lopinavir, and favipiravir). This combination among many others are the final step of this computational work, before they are presented to domain experts for interpretation.

The depiction of Figure 3 explains the general idea with an example of two cliques, each consisting of four components. Here, we fully discuss all the cliques resulted from this analysis, which we also list in the Results Section 3 above (Table 4). The main purpose of this step is to validate the drug combination extracted from the biomedical literataure. We start by listing the findings in groups of three rows: (a) findings from the publications, (b) findings from clincal trai records, and (3) the matching components. Table 6 presents ten groups of cliques and a match of three-drug combinations to be discusse onward.

From the ten groups of cliques listed in Table 6, we observed four unique matching combinations. Here, we discuss each of those four and provide further insights.

The three components *hydroxychloroquine*, *azithromycin*, and *doxycycline* match between four pairs of size-four cliques. This is due to the fact that the combination of hydroxychloroquine and azithromycin is possible and also the combination of hydroxychloroquine and doxycycline is possible, as indicated in Table 5. The evidence of using hydroxychloroquine and azithromycin is reported from the clinical trials [70,71,72,73]. Moreover, both hydroxychloroquine and ciprofloxacin doxycycline were commonly studied in clinical trials [74], where the results did not seem impressive and further investigations were recommended. There is no evidence that a combination of azithromycin and doxycycline has been investigated for COVID-19 treatment, although azithromycin may be used as an alternative to doxycycline for other infections (e.g., urogenital Chlamydia trachomatis infection) [75].The three components *hydroxychloroquine*, *ritonavir*, and *favipiravir* match between two pairs of size-four cliques. In Table 2 we reported the combination of hydroxychloroquine and favipiravir. This combination is particularly used as a home treatment of older people who are COVID-19 symptomatic. This recommendation for treatment is based on clinical trials that are still recruiting participants [76]. There was no evidence of combining ritonavir with favipiravir. There was, however, a comparison of efficacy between the two drugs, and the study concluded that “Favipiravir does not reduce the number of ICU admissions or intubations or in-hospital mortality” [77]. It is important to note that both lopinavir and ritonavir are used interchangeably due to the fact that they are both sold under the brand name Kaletra and, therefore the cliques that contain one also contain the other.The three components *hydroxychloroquine*, *azithromycin*, and *ivermectin* match between two pairs of size-four cliques. A combination of hydroxychloroquine and azithromycin was discussed above. Azithromycin and ivermectin are new components in these cliques. Both are co-administered for other conditions (scabies and impetigo) [78]. Recently, a study recommended adding ivermectin as a solution to the COVID-19 treatment protocol that combines hydroxychloroquine, favipiravir, and azithromycin [79].The three components *hydroxychloroquine*, *lopinavir*, and *ritonavir*. Both hydroxychloroquine and lopinavir have been explained in the second item of this list. As for lopinavir and ritonavir, they both explained earlier and commercialized under the brand name Kaltera.

The discussion above showed an itemized list of the drug combinations extracted fom the biomedical literature, the overlapping combination found in clinical trials, and the individual matching drugs. Specifically, it included two-drug combination in Table 5 indicated that seven drug pairs are possibe to combine without significant clinical impediments. Among these, two are already in use. For the three or more drug combinations (Table 4), several three-way combinations could be identified as potentially worthwhile to investigate.

### 4.3. Clinical Trials Supporting Evidence and Stats

Here, we provide more supporting evidence to the finding of this work. Here, we list the up-to-date statistics gathered from the clinical trials that investigting the drug combinations, as predicted ealier in this paper. The columns of Table 7 show drug ombinations, in pairs and triples, their current clinical trial status, and the number of trials study each pair or triple. The table captures the follows (recruiting, completed, active and not recruiting, not recruiting yet) whenever available. Two specific pairs are worthy noting: (1) the most studied combinations is ritonavir and lopinavir (which is another supporting evidence of the significance of ritonavir). Retonavir remains among the most promising component among the emerging treatments e.g., (Kaltera or Paxlovid). (2) The least studied combination is ruxolitinib and colchicine despite that both studies are still recruiting. It is unclear whether the low frequencey is due to novelty of this treatment or any other factor.

#### Limitations

Our analysis hinges on the literature collection that is used as the basis for inferring co-occurrences of drug mentions. In this work, we adopted a simple strategy for collecting COVID-19 related literature. Other datasets with more comprehensive coverage are now available, including the CORD-19 dataset [80] or LitCovid [81]. More comprehensive lists for COVID search terms are now available [82]. We leave exploration of the use of these resources for future work. Another limitation is that this analysis did not predict the potential therapy for severe versus non-severe COVID cases.

## 5. Conclusions

In this paper, we tested the hypothesis that a combination of FDA-approved drugs may be considered as a COVID-19 candidate treatment if there exists (1) evidence in the COVID-19-related biomedical literature suggesting the combination and (2) an evidence from clinical trials that provides grounds for this combination. We also introduced our computational framework for COVID-19 drug repurposing, which is centered around analyzing literature-based networks for clique patterns. The findings were compared with evidence mined from clinial for validation. Indeed, the hypothesis was validated algorithmically and it has proven to be not only valid hypothesis but also a promising drug repurposing prediction tool. We further validated the clique pattens by domain experts and classified into different combineability categories. The investigation provided adequate explanations based on the publicly available data. We believe the reported FDA-approved drug combinations (either in pairs or triples) are convincing and promising as COVID-19 treatments, especially those already commercialized (e.g., ritonavir-boosted nirmatrelvir).

Notably, the potential COVID-19 treatments reported in this paper are entirely based on the COVID-19 biomedical publications and ongoing COVID-19 clinical trials. Thus, the findings are subject to further investigations, unless already commercialized as in the case for kaletra. Specifically, the authors will identify additional validation sources to advance the understandings of the cliques that did not have coverage from the clinical trials records. Additionally, the investigations will pay a special focus on how such drug combination may affect the treatment of patients with preexisting conditions like asthma, depression, diabetes, hypertension, etc.

## Figures and Tables

**Figure 1 pharmaceutics-14-00567-f001:**
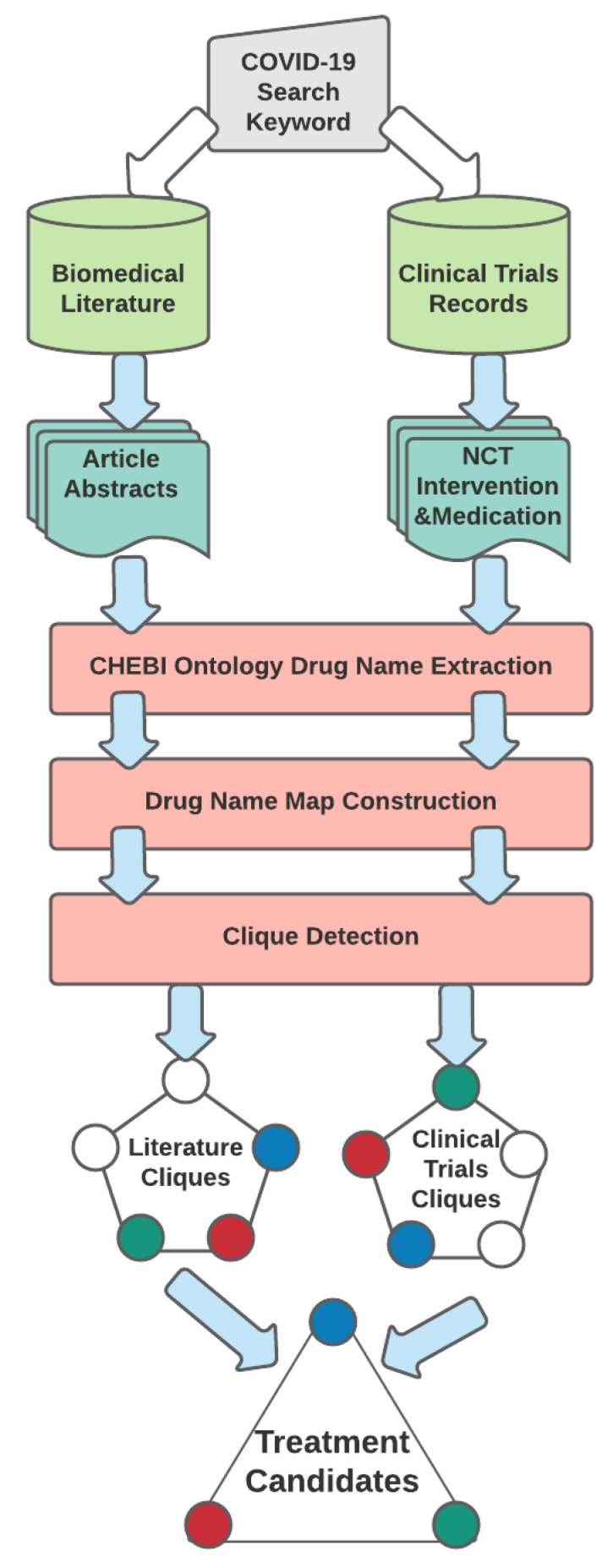
Workflow of the framework starting with the raw inputs (biomedical publications and clinical trial intervention and medication recrods). There are three steps: (1) ChEBI ontology term extraction, (2) Network construction from the associated drugs, and (3) applying a clique finding algorithm to detect the cliques. The final step is the discovery of comparing and matching cliques found in the two sources and compared for similarities. The intersection consititutes a candidate COVID-19 treatment.

**Figure 2 pharmaceutics-14-00567-f002:**
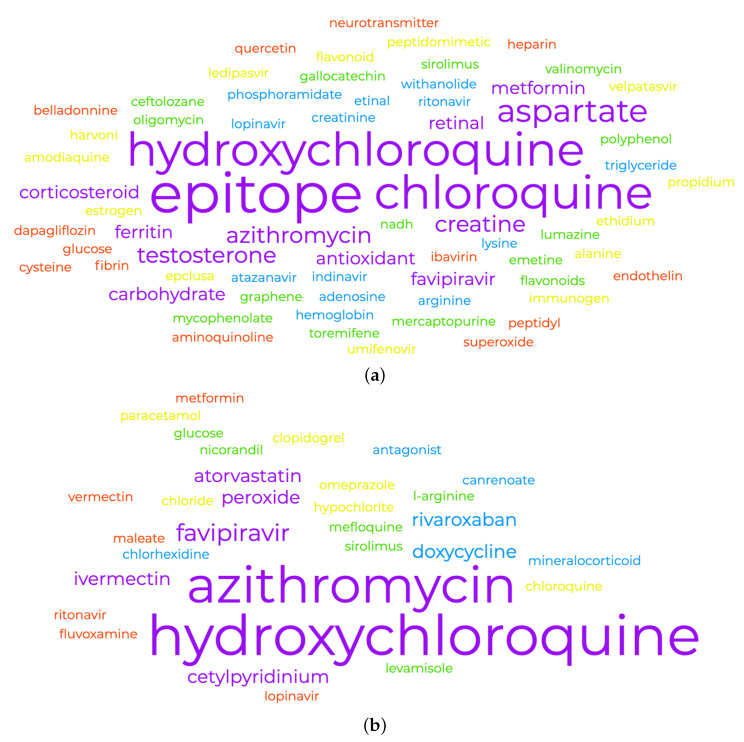
(**a**) Drugs and chemical entities extracted from the abstracts of the biomedical publications studying COVID-19. (**b**) Drugs and chemical entities extracted from the intervention and treatment sections of clinical trials investigating COVID-19. The list of drugs extracted from the biomedical publication (**a**) and the clinical trial records (**b**). Overlap of key drugs (ritonavir, lopinavir, favipiravir, hydroxychloroquine, chloroquine, and azithromycin) between the two affirms the validity of our hypothesis.

**Figure 3 pharmaceutics-14-00567-f003:**
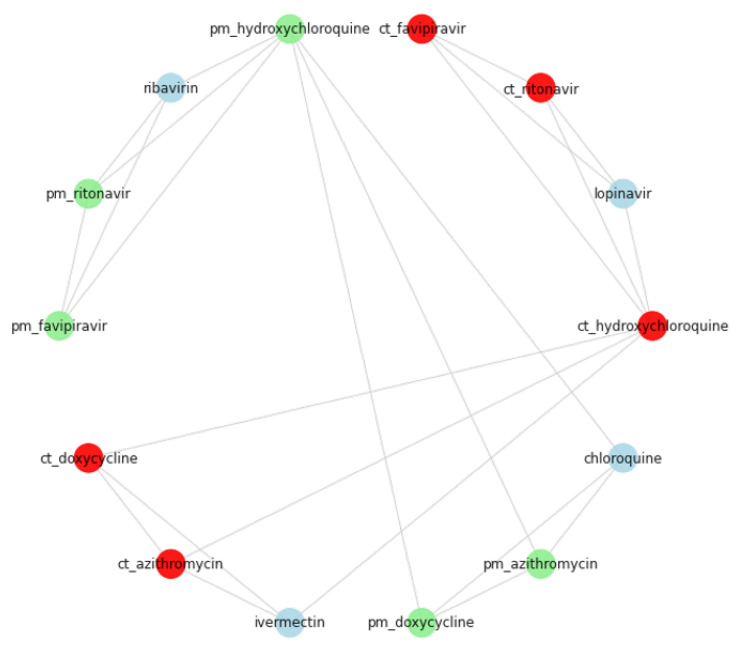
A map of size-four literature cliques and clinical trial cliques resulting from the workflow. The green nodes are the components of cliques detected from the biomedcial publications; they are prefixed with “pm”. The red nodes are the matching counterparts extracted from the clincical trial datasets; they are prefixed with “ct”. The lightblue nodes are part of the clique but do not have a match and, therefore, cannot be part of the result.

**Table 1 pharmaceutics-14-00567-t001:** The raw result of analyzing the networks constructed from the biomedical publications for clique. The columns show the 10-folds that made up the dataset and the score. The rows show the size of the cliques. The last row shows the total number of cliques detected from each fold. Overall, the total number of all cliques detected from the publication is 3643.

–	F1	F2	F3	F4	F5	F6	F7	F8	F9	F10
Clique size: 2	120	135	118	296	139	132	126	126	141	126
Clique size: 3	101	134	169	163	137	136	132	132	45	43
Clique size: 4	68	81	97	130	86	53	67	67	0	9
Clique size: 5	32	27	78	108	54	0	16	16	1	2
Total	321	377	462	697	416	321	341	341	187	180

**Table 2 pharmaceutics-14-00567-t002:** Cliques of size two (drugs qualified with their ChEBI ID) detected from clinical trials association network.

Drug 1	Drug 2
ChEBI:63608_maraviroc	ChEBI:134722_favipiravir
ChEBI:6970_mometasone	ChEBI:50858_corticosteroid
ChEBI:72291_cobicistat	ChEBI:367163_darunavir
ChEBI:28775_hesperidin	ChEBI:4631_diosmin
ChEBI:85973_edoxaban	ChEBI:28304_heparin
ChEBI:85973_edoxaban	ChEBI23359_colchicine
ChEBI:85089_ledipasvir	ChEBI:85083_sofosbuvir
ChEBI:23965_estradiol	ChEBI:17026_progesterone
ChEBI:23965_estradiol	ChEBI:50114_estrogen
ChEBI:6015_isoflurane	ChEBI:9130_sevoflurane

**Table 3 pharmaceutics-14-00567-t003:** Cliques of size three detected from clinical trials association network.

Drug 1	Drug 2	Drug 3
peroxide	cetylpyridinium	chlorhexidine
l-arginine	atorvastatin	nicorandil
hydroxychloroquine	azithromycin	mefloquine
hydroxychloroquine	azithromycin	favipiravir
hydroxychloroquine	azithromycin	glucose
hydroxychloroquine	azithromycin	sirolimus
hydroxychloroquine	azithromycin	levamisole
hydroxychloroquine	chloroquine	favipiravir
paracetamol	ivermectin	azithromycin

**Table 4 pharmaceutics-14-00567-t004:** Cliques of size four detected from clinical trials association network.

Drug 1	Drug 2	Drug 3	Drug 4
omeprazole	rivaroxaban	clopidogrel	atorvastatin
hydroxychloroquine	lopinavir	ritonavir	favipiravir
hydroxychloroquine	azithromycin	doxycycline	ivermectin
hydroxychloroquine	azithromycin	doxycycline	rivaroxaban

**Table 5 pharmaceutics-14-00567-t005:** Two-drug combinations of COVID-19 treatment candidates identified for further investigation.

Drug 1	Drug 2	Combineability
Estrogen (ChEBI:50114)	Estradiol (ChEBI:23965)	No
Hydroxyethylidene(ChEBI:5801)	Azithromycin (ChEBI:2955)	Possible
Lopinavir (ChEBI:31781)	Ritonavir (ChEBI:45409)	Yes
Ruxolitinib(ChEBI:66919)	Colchicine (ChEBI:23359)	Possible
Hydroxychloroquine (ChEBI:5801)	Favipiravir ChEBI:134722	Possible
Hydroxychloroquine (ChEBI:5801)	Chloroquine ChEBI:3638	No
Azithromycin (ChEBI:2955)	Ivermectin ChEBI:6078	Possible
Hydroxychloroquine (ChEBI:5801)	Lopinavir(ChEBI:31781)	Probably not
Hydroxychloroquine (ChEBI:5801)	Doxycycline(ChEBI:50845)	Possible
Daclatasvir (ChEBI:82977)	Sofosbuvir(ChEBI:85083)	Yes

**Table 6 pharmaceutics-14-00567-t006:** Cliques of size four from the biomedical publications and clinical trial records that have three components in common.

Source	Clique Components
Biomed pub	hydroxychloroquine, chloroquine, azithromycin, doxycycline
Clinical trial	hydroxychloroquine, azithromycin, doxycycline, ivermectin
Match	hydroxychloroquine, azithromycin, doxycycline
Biomed pub	hydroxychloroquine, mycophenolate, azithromycin, doxycycline
Clinical trial	hydroxychloroquine, azithromycin, doxycycline, ivermectin
Match	hydroxychloroquine, azithromycin, doxycycline
Biomed pub	hydroxychloroquine, chloroquine, azithromycin, doxycycline
Clinical trial	hydroxychloroquine, azithromycin, doxycycline, rivaroxaban
Match	hydroxychloroquine, azithromycin, doxycycline
Biomed pub	hydroxychloroquine, mycophenolate, azithromycin, doxycycline
Clinical trial	hydroxychloroquine, azithromycin, doxycycline, rivaroxaban
Match	hydroxychloroquine, azithromycin, doxycycline
Biomed pub	hydroxychloroquine, oseltamivir, ritonavir, favipiravir
Clinical trial	hydroxychloroquine, lopinavir, ritonavir, favipiravir
Match	hydroxychloroquine, ritonavir, favipiravir
Biomed pub	hydroxychloroquine, ribavirin, ritonavir, favipiravir
Clinical trial	hydroxychloroquine, lopinavir, ritonavir, favipiravir
Match	hydroxychloroquine, ritonavir, favipiravir
Biomed pub	hydroxychloroquine, azithromycin, macrolide, ivermectin
Clinical trial	hydroxychloroquine, azithromycin, doxycycline, ivermectin
Match	hydroxychloroquine, azithromycin, ivermectin
Biomed pub	hydroxychloroquine, azithromycin, ivermectin, ritonavir
Clinical trial	hydroxychloroquine, azithromycin, doxycycline, ivermectin
Match	hydroxychloroquine, azithromycin, ivermectin
Biomed pub	hydroxychloroquine, darunavir, lopinavir, favipiravir
Clinical trial	hydroxychloroquine, lopinavir, ritonavir, favipiravir
Match	hydroxychloroquine, lopinavir, favipiravir
Biomed pub	hydroxychloroquine, ligand, lopinavir, ritonavir
Clinical trial	hydroxychloroquine, lopinavir, ritonavir, favipiravir
Match	hydroxychloroquine, lopinavir, ritonavir

**Table 7 pharmaceutics-14-00567-t007:** Two or three drug combinations in clinical trials that confirm the findings from the biomedical literature.

Drug Combination	Clinical Trial Status	# Trials
ritonavir and lopinavir	Recruiting	20
ritonavir and lopinavir	Completed	15
ritonavir and lopinavir	Active, not recruiting	7
ruxolitinib and colchicine	Recruiting	2
hydroxychloroquine and favipiravir	Completed	13
hydroxychloroquine and favipiravir	Recruiting	2
hydroxychloroquine and favipiravir	Active, not recruiting	3
azithromycin and ivermectin	Completed	6
azithromycin and ivermectin	Recruiting	6
hydroxychloroquine and lopinavir	Recruiting	18
hydroxychloroquine and lopinavir	Completed	12
hydroxychloroquine and lopinavir	Active, not recruiting	5
hydroxychloroquine and doxycycline	Completed	4
hydroxychloroquine and doxycycline	Recruiting	1
daclatasvir and sofosbuvir	Recruiting	5
daclatasvir and sofosbuvir	Completed	3
daclatasvir and sofosbuvir	Not recruiting yet	1
hydroxychloroquine and chloroquine	Recruiting	8
hydroxychloroquine and chloroquine	Completed	16
hydroxychloroquine and chloroquine	Active, not recruiting	5
hydroxychloroquine, lopinavir, ritonavir	Recruiting	3
hydroxychloroquine, lopinavir, ritonavir	Recruiting	9
hydroxychloroquine, lopinavir, ritonavir	Active, not recruiting	3
hydroxychloroquine, lopinavir, ritonavir	Not recruiting yet	3
hydroxychloroquine, lopinavir, favipiravir	Recruiting	1
hydroxychloroquine, lopinavir, favipiravir	Completed	5
hydroxychloroquine, lopinavir, favipiravir	Active, not recruiting	1
hydroxychloroquine, azithromycin, ivermectin	Completed	4
hydroxychloroquine, azithromycin, ivermectin	Active, not recruiting	1
hydroxychloroquine, ritonavir, favipiravir	Recruiting	1
hydroxychloroquine, ritonavir, favipiravir	Completed	5
hydroxychloroquine, ritonavir, favipiravir	Active, not recruiting	1
hydroxychloroquine, azithromycin, doxycycline	Recruiting	1
hydroxychloroquine, azithromycin, doxycycline	Completed	3
hydroxychloroquine, azithromycin, doxycycline	Not recruiting yet	1

## Data Availability

The results of this paper is entirely documented in the Tables of this paper.

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
