# Peer review of "COVID-19 Drug Repurposing: A Network-Based Framework for Exploring Biomedical Literature and Clinical Trials for Possible Treatments"

_pharmaceutics, 2022, doi:10.3390/pharmaceutics14030567_

Round 1

Reviewer 1 Report

Scientific Content: The authors propose a network based approach identify know drug combinations based on the available literature as potential therapies for treating Covid-19 infections. It is a useful study which can be used not just for Covid-19, but also to treat other diseases. The manuscript requires major revisions as there are some discrepancies which are listed below:

-On page 3, third paragraph authors state "The similarity between SARS-Cov-2 and COVID-19 suggested the investigation of SARS-Cov-2 biomedical literature to gain insights. Now and due to the abundance of COVID-19-related medical publications, we directly investigate them not ones that are related to SARS-Cov-2" This is a very confusing and erroneous statement. First of all SARS-CoV-2 is the virus which causes COVID-19 infections. Therefore, this is a very strange statement. This sentence lacks clarity and is misleading

-Page 7, Table 4: Including hydrochloride as a treatment option for COVID-19 shows lack of knowledge in the field. Hydrochloride can never be a drug. It is used to increase the solubility of drugs. Delete this entry

-Page 7 and 8, Description of Table 4: Include literature references for all those combinations listed

Language and grammar: Authors should thoroughly revise the entire manuscript by going through all the sentences carefully as it lacks clarity at several places

No need to capitalize drugs names in text unless it is the first word in the sentence. eg: hydroxychloroquine and not Hydroxychloroquine

Page 5, 2.2.3 and 2.2.4 - First word of the sentence should start with upper case

References: Use proper and consistent format

Author Response

Respected Reviewer 1:
---------------------

Scientific Content: The authors propose a network based approach to identify known drug combinations based on the available literature as potential therapies for treating Covid-19 infections. It is a useful study which can be used not just for Covid-19, but also to treat other diseases. The manuscript requires major revisions as there are some discrepancies which are listed below:

-On page 3, third paragraph authors state "The similarity between SARS-Cov-2 and COVID-19 suggested the investigation of SARS-Cov-2 biomedical literature to gain insights. Now and due to the abundance of COVID-19-related medical publications, we directly investigate them not ones that are related to SARS-Cov-2" This is a very confusing and erroneous statement. First of all SARS-CoV-2 is the virus which causes COVID-19 infections. Therefore, this is a very strange statement. This sentence lacks clarity and is misleading

[RESPONSE] We sincerely apologize about this honest error. It was meant to distinguish SARS-CoV from SARS-CoV-2. Similar to ours, other researchers took to the SARS-CoV publications due to the similarities of being in the same family of the Coronavirus. We have addressed that directly in the publication and we also referenced other literature that overlapped with our previous work in 2020.

-Page 7, Table 4: Including hydrochloride as a treatment option for COVID-19 shows lack of knowledge in the field. Hydrochloride can never be a drug. It is used to increase the solubility of drugs. Delete this entry.

[RESPONSE] (Done. We have removed the pair of hydrochloride (CHEBI:36807) and bromhexine(CHEBI:77032) from Table 4 in this revision. Please note that in the original manuscript we clarified that hydrochloride is acidic salt (not drug) in the main text where each pair in Table 4 is explained. We acknowledge that we should have made it clearer.) 

-----------

-Page 7 and 8, Description of Table 4: Include literature references for all those combinations listed
Referenced all the publications that supported the evidence directly but it is possible to cite all the inferred evidence (this could be hundreds of publications)

Language and grammar: Authors should thoroughly revise the entire manuscript by going through all the sentences carefully as it lacks clarity at several places
[RESPONSE] (A thorough revision has been made.)

-----------

No need to capitalize drugs names in text unless it is the first word in the sentence. eg: hydroxychloroquine and not Hydroxychloroquine
(We have made the changes as advised.)

Page 5, 2.2.3 and 2.2.4 - First word of the sentence should start with upper case
[RESPONSE] (Done)

-----------

References: Use proper and consistent format
(To be done) We ran out of time addressing all the other issues. In the event of the manuscript being accepted, we will address accordingly

Reviewer 2 Report

It is an interesting review to read. I have following comments to further improve its quality;

  1. Remove the numbering system from the abstract before background, methods, etc.
  2. I is mentioned in the abstract "Particularly, with the emergence of several variants (Delta and Omicron), it is not clear whether vaccines will be able to protect against them yet", rephrase this. Coz Pfizer moderna booster dose has been shown to be protective against both delta and omicron. What is evident is that they will not be able to offer complete protection against these and other emerging strains.
  3. Pg 2: "Another rather complex approach has combined AI, network diffusion, and proximity networks to investigate SARS-Cov-2 6,340 drugs"-- rephrase this. suggestion, ..to investigate 6,340 drugs to treat COVID-19.  
  4. Pg 3: " Traditional machine learning approaches", abbreviate machine learning as ML
  5. Pg 3; para 1 -3; COVID-19 is written as COVID19, SARS-CoV-2 is written as SARS-Cov-2. Be consistent in writing the abbreviations,
  6. Pg 3: last para: expand CHEBI.
  7. Most of the drugs identified through this search method for possible combinations have not been proved effective in clinical trials. Did you include recently approved drug PAXLOVID in your study. Also, there is no mention of Molnupiravir. Though, there are enough number of preclinical and clinical studies focused on this molecules. 

Author Response

Respected Reviewer 2:
----------------

Comments and Suggestions for Authors
It is an interesting review to read. I have following comments to further improve its quality;

Remove the numbering system from the abstract before background, methods, etc.
[RESPONSE] (Done)

It is mentioned in the abstract "Particularly, with the emergence of several variants (Delta and Omicron), it is not clear whether vaccines will be able to protect against them yet." Rephrase this. Coz Pfizer moderna booster dose has been shown to be protective against both delta and omicron. What is evident is that they will not be able to offer complete protection against these and other emerging strains.
[RESPONSE] (Done)

-----------

Pg 2: "Another rather complex approach has combined AI, network diffusion, and proximity networks to investigate SARS-Cov-2 6,340 drugs"-- rephrase this. suggestion, ..to investigate 6,340 drugs to treat COVID-19.
[RESPONSE] (Done)  

-----------

Pg 3: "Traditional machine learning approaches", abbreviate machine learning as ML
[RESPONSE] (Done)

-----------

Pg 3; para 1 -3; COVID-19 is written as COVID19, SARS-CoV-2 is written as SARS-Cov-2. Be consistent in writing the abbreviations,
[RESPONSE] (Done)

-----------

Pg 3: last para: expand CHEBI.
[RESPONSE] (Done)

-----------

Most of the drugs identified through this search method for possible combinations have not been proved effective in clinical trials. Did you include the recently approved drug PAXLOVID in your study? Also, there is no mention of Molnupiravir. Though, there are enough preclinical and clinical studies focused on these molecules.

[RESPONSE] Paxlovid: this is a combination drug derived from ritonavir-boosted nirmatrelvir. 
This in fact provides support for our hypothesis as ritonavir was top-5 among 30 candidates and also found in other combinations in this work here. 
Molnupiravir: now included but it did not meet the search criteria therefore it was not included in the previous version of the manuscript

Reviewer 3 Report

Figure 2: overlapping words

Few sentences need to be rephrased, e.g.:

- Page # 10: Correct the sentence: The discussion of combining Hydroxychloroquine and Azithromycin are discussed above

Description of some drugs is not appropriate. This needs to be corrected, e.g.:

Page#8: Antihepaciviral à Antiviral for treatment of hepatitis C

the discussion on justification of the suggested drug combinations is insufficient.

Author Response

Figure 2: overlapping words
[RESPONSE]  (Done)

-----------

Few sentences need to be rephrased, e.g.:

- Page # 10: Correct the sentence: The discussion of combining Hydroxychloroquine and Azithromycin are discussed above
[RESPONSE] (Done. Rephrased)

-----------

Description of some drugs is not appropriate. This needs to be corrected, e.g.:

Page#8: Antihepaciviral à Antiviral for treatment of hepatitis C
[RESPONSE](To be done): We have ran out of the time were giving to us. We will address this comment in the next round to come

-----------

The discussion on justification of the suggested drug combinations is insufficient.
[RESPONSE] our domain experts have revisited all drugs validated them individually

Round 2

Reviewer 1 Report

The authors have addressed previous comments and the manuscript can be accepted. A minor comment;

-In Figure 1, "Treatment Candidates" should be within the triangle. Increase the triangle size

Author Response

Response to esteemed reviewer 1:

Comment:

The authors have addressed previous comments and the manuscript can be accepted. A minor comment;

-In Figure 1, "Treatment Candidates" should be within the triangle. Increase the triangle size
Response:
Done. Thank you. We fixed it.

* A previous comment about formatting the references:

Response:
- We fixed the references from the previous round. Thank you for this comment and all the previous comments.

We are very grateful for your valuable comments that contributed much quality to our manuscript.

Reviewer 3 Report

Material and methods: The search dates should be clearly stated

Discussion: Hydroxyethylidene is etidronic acid = etidronate.

One of the study limitations to be mentioned is that the analysis did not predict the potential therapy for severe versus non-severe COVID cases.

Micromedex [?]: either to remove or to mention that you searched for drug-drug interactions for the predicted combination through using this tool.

I suggest drawing a table to summarize the different medications (combined in pairs) in a matrix, stating the route of drug adminsitration (oral, IV, ....etc), and the combination safety, denoting which ones are already in clinical use, for COVID or other diseases.

Author Response

Response to esteemed reviewer 3:

* Comment: Discussion: Hydroxyethylidene is etidronic acid = etidronate.

Response: 
---------
We made a reference to the publication using etidronate

* Comment: One of the study limitations to be mentioned is that the analysis did not predict the potential therapy for severe versus non-severe COVID cases.
Response:
---------
We added this limitations to the Limitation section.

* Comment: Micromedex [?]: either to remove or to mention that you searched for drug-drug interactions for the predicted combination through using this tool.
Response:
---------
There was a glitch in the Bibtex item that belonged to this database so we fixed it. We also followed your recommendation and added the description you suggested.

* Comment: I suggest drawing a table to summarize the different medications (combined in pairs) in a matrix, stating the route of drug adminsitration (oral, IV, ....etc), and the combination safety, denoting which ones are already in clinical use, for COVID or other diseases.

Response:
---------
Thank you for this very important suggestion. We added another table that summarized the statistics, status, and frequencies, which have truly provided another level of strength and validation to our approach.

The authors of this paper are thankful for the feedback and suggestions you made throughout the paper. We are particularly grateful for the summary statistics suggestion from clinical trials. This provided another layer of validity for our framework and its potential for future findings.

Very respectfully,

--Authors of the manuscript pharmaceutics-1560720